# Role of literacy, fear and hesitancy on acceptance of COVID-19 vaccine among village health volunteers in Thailand

**Pallop Siewchaisakul**[1,2], **Pongdech Sarakarn**[3,4], **Sirinya Nanthanangkul**[5], **Jirapat Longkul**[6], **Waraporn Boonchieng**[1,2], **Jukkrit Wungrath**[1]*

1 Faculty of Public Health, Chiang Mai University, Chiang Mai, Thailand, 2 The Center of Excellence in Community Health Informatics, Chiang Mai University, Chiang Mai, Thailand, 3 Epidemiology and Biostatistics Department, Faculty of Public Health, Khon Kaen University, Khon Kaen, Thailand, 4 ASEAN Cancer Epidemiology and Prevention Research Group, Khon Kaen, Thailand, 5 Research Publishing and Academic support Department, Udonthani Cancer Hospital, Department of Medical Services, Ministry of Public Health, Nong Phai, Thailand, 6 Faculty of Public health, Thammasat University, Bangkok, Thailand

* jukkrit.w@cmu.ac.th

**Data Availability Statement:** All relevant data are within the article and its Supporting information files.

## Abstract

### Background

The roles of literacy, fear and hesitancy were investigated for acceptance of COVID-19 vaccine (AV) types among village health volunteers (VHVs) in Thailand.

### Materials and methods

A cross-sectional study was conducted using an unidentified online questionnaire to assess literacy, fear and hesitancy of COVID-19 vaccine acceptance among Thai VHVs between 1 and 15 October 2021. The questionnaire was developed based on the HLVa-IT (Health Literacy Vaccinale degli adulti in Italiano) for vaccine literacy (VL), using an adult Vaccine Hesitancy Scale (aVHS) for COVID-19 vaccine hesitancy (VH) and Fear of COVID-19 scale (FCoV-19S) for the distress of COVID-19 vaccine. The effects of VL, VH and vaccine fear (VF) on AV were estimated using multivariable logistic regression.

### Results

A total of 5,312 VHVs completed the questionnaire. After adjustment with variables in the multivariable analysis, the VL score was insignificantly associated with increased vaccination (aOR = 1.002; (95%CI: 0.994–1.01)), while VF and VH significantly decreased the chance of vaccination, aOR = 0.966 (95%CI: 0.953–0.978) and aOR = 0.969; (95%CI: 0.960–0.979), respectively and VF and VH were negatively associated with AV for all types of vaccine preference, with VL showing a reverse relationship only for mRNA-based vaccines.

**Funding:** This study was granted by the Center of Excellence in Community Health Informatics, Chiang Mai University, Chiang Mai, Thailand.

**Competing interests:** The authors have declared that no competing interests exist.

## Conclusion

VL may not increase AV among VHVs. To increase attitudes toward receiving COVID-19 vaccination in Thailand, the government and health-related organizations should instigate policies to significantly reduce VF and VH among Thai VHVs.

## Introduction

Coronavirus disease 2019 (COVID-19) is a communicable sickness caused by the severe acute respiratory syndrome coronavirus 2 (SARS-CoV-2) [1]. COVID-19 was officially declared widespread by the World Health Organization (WHO) on 11 March 2020 [2]. Case numbers and deaths from the disease are still increasing globally [3]. COVID-19 vaccines have shown promise as a prophylactic measure for protection against infection, preventing severe symptoms and slowing the rapid spread of the disease [4–6]. As of conducting research in October 2021, global vaccination coverage was 34%, with 23% in Thailand, far below the level of herd immunity [7]. The target of vaccination in Thailand is 70% [8]. However, current vaccination coverage in May 2022 is 70.3% [7].

Previous studies reported that the acceptance rate of the COVID-19 vaccine varied by countries and different time points [9]. Vaccine acceptance rate was 37.40% in Jordan, 61.16% in Bangladesh, 56.90% in the EU, 80.00% in the USA and 63% in Africa [10–14]. Known factors contributing to the acceptance of vaccine (AV) were vaccine literacy (VL), vaccine hesitancy (VH) and vaccine fear (VF). VH and VF had a negative impact on AV, while VL [15, 16] showed a positive impact [17, 18]. Low health literacy impacts VH and may result in refusal or delay in AV [19]. A study among French adults showed that high health literacy scores were associated with the intention to get vaccinated with minimal VH [17], while U.S. college students showed higher score, were positively associated with greater willingness for COVID-19 vaccination [20]. Rapid transmission of the COVID-19 pandemic has increased the fear of virus transmission in the community. A recent study in Vietnam demonstrated that health literacy modified the effect of fear; however, on quality of life [21].

Sirikalyanpaiboon et al. conducted a survey among Thai physicians. They found that preference for particular vaccines was independently associated with VH, especially for the mRNA vaccine [22]. Another study surveyed the general population and reported the AV rate at 41.8%. As well as adenovirus-based and mRNA-based vaccines, an inactive vaccine type is also available in Thailand. Acceptance rate increased from 89.0% to 91.3% if people could select the vaccine brand and 80.7% to 83.2% for brands recommended by healthcare professionals [23].

To combat the outbreak of COVID-19 in Thailand, the Ministry of Public Health recruited 1.04 million village health volunteers (VHVs) throughout the nation to help contain the spread of the disease. The emphasis was on humanizing and updating people about the cause, prevention and treatment of diseases. The first COVID-19 vaccine arrived in Thailand in February 2021 and the Thai Prime Minister encouraged VHVs to boost public confidence in this vaccine. Later, novel variants of COVID-19 and new vaccines arrived in Thailand. The VHVs assisted in communicating vaccine information to the public [24, 25] and encouraged the acceptance of the vaccine as a positive way to reduce the spread of the disease.

As aforementioned, few studies have examined the relationship between VL, VF and VH on AV, particularly by vaccine types, while no studies have been conducted in Thailand, especially among VHVs who are at the frontline of all community health matters and influence vaccine perceptions of community members. Therefore, this research investigated the consequences of VL, VF and VH on AV among VHVs in Thailand.

## Research methodology

### Study design and settings

We conducted a cross-sectional study between 1 and 15 October 2021 via the online platform of VHVs in Thailand.

### Study samples and data collection

Eligible participants in this study were VHVs aged over 18 and registered in the mobile application SMART VHV. The total number of Thai registered mobile application SMART VHVs aged over 18 in 2021 was 254,743 people [26], with 137,782 records available for contact. A nonprobability snowball sampling method was adopted based on the 137,782 registered SMART VHVs through the social platforms Line and Facebook as the two most popular social media platforms in Thailand and used by VHVs to communicate and coordinate with each other.

First, we asked for cooperation with VHV leaders according to their responsible health areas. We uploaded our questionnaire to their social media and the representative VHV leaders forwarded the questionnaire to Line or Facebook groups for completion by their VHVs. The study subjects were screened for inclusion criteria including age and registration of mobile application SMART VHV. Eligible participants were asked to sign an informed consent form by clicking on it. After signing the informed consent, the structured questionnaire appeared on their screens. The participants completed and submitted the questionnaire via the online platform Google Form. They were free to withdraw at any time and the survey took approximately 10 minutes to complete. All completed questionnaire was stored via Google Form. SMART VHVs who lack of address information and incomplete filling questionnaire were excluded. A total of 5,312 VHVs responded to our survey (Fig 1).

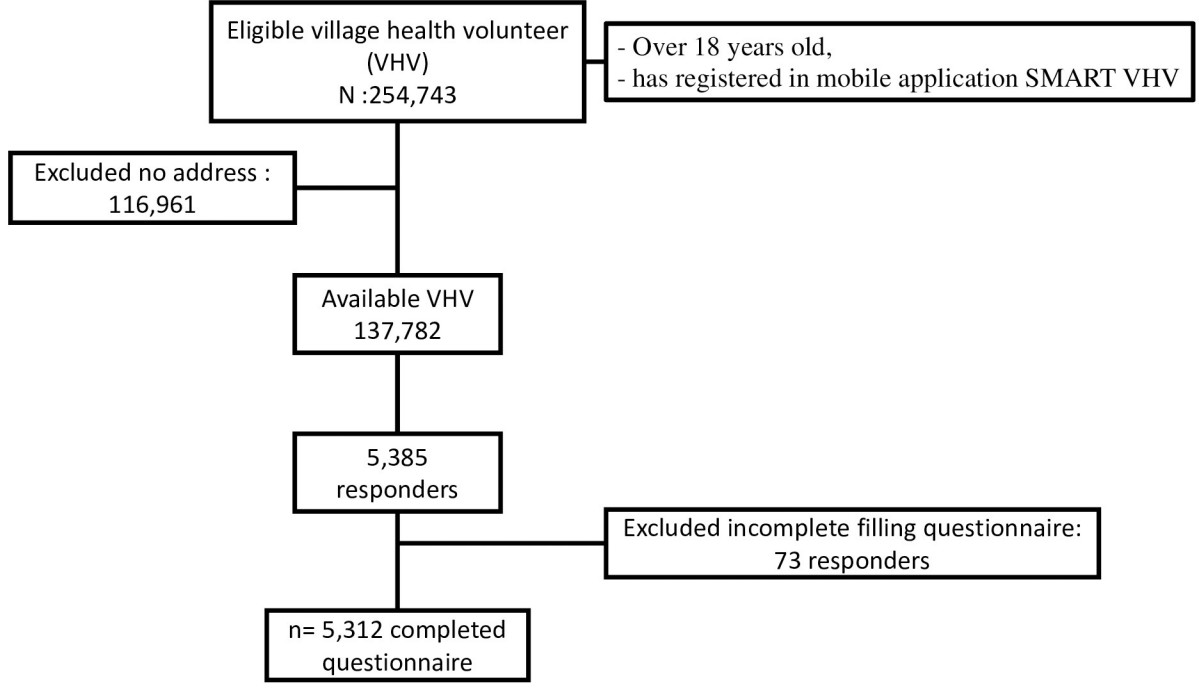

**Fig 1. Flow chart of sample collection.**

## Instruments and measurements

The questionnaire was designed and pretested by the researchers after an extensive literature review. Our questionnaire was self-rated and asked about demographic characteristics, VL, VF, VH and AV. Details of instruments, tools and measurements are presented below.

## Demographic data

Demographics included the following data: sex, age, educational level, marital status, religion, occupation, medical condition, family income and work experience.

## Vaccine literacy (VL)

A self-rated VL questionnaire including three domains as functional, communicative and critical was developed. The VL questionnaire was adapted from the Health Literacy Vaccinale degli adulti in Italiano (HLVa-IT) [27]. The VL questionnaire was composed of 12 items (questions) including functional VL (items L1 to L5), for example, "I did more research on the COVID-19 vaccine", "I know where to find information on the COVID-19 vaccine", communicative VL (items L6 to L10), for example, "I have consulted or received advice regarding the COVID-19 vaccine from a doctor, nurse or healthcare professional", "I can analyze the quality or accuracy of the information I find on the COVID-19 vaccine" and critical VL (items L11 to L12), for example, "I pass on the correct information about the COVID-19 vaccine to others". Answers were supplied by the interviewees according to a Likert scale with four possible choices (4-never, 3-rarely, 2-sometimes and 1-often). Scores were calculated using the mean value of each scale response ranging from 1 to 4, with higher values indicating greater VL.

## Vaccine fear (VF)

A 5-item questionnaire was conducted to estimate the fear of COVID-19 vaccine, with a four-point Likert scale used to examine whether or not people were apprehensive of vaccination, ranging from 1 to 4 as 4-never, 3-rarely, 2-sometimes and 1-often. Examples of items included, "I am very afraid of having to get vaccinated against COVID-19", and "I am afraid of dying from vaccination against COVID-19". The aggregate score was the total scores of the 5 items ranging from 5 to 20, with a higher total demonstrating greater fear of the COVID-19 vaccine.

## Vaccine hesitancy (VH)

Our VH questionnaire was adapted from Akel Kb et al. as the so-called Hesitancy Scale (aVHS) [28]. The VH had 7 items with a five-point Likert scale as answer choices, ranging from 1 to 5 as 5-strongly disagree, 4-disagree, 3-neutral, 2-agree and 1-strongly agree. Examples of the items included "The COVID-19 vaccine is very important to me", and "The COVID-19 vaccine currently in use in Thailand is effective". The aggregate score was the total of each item score ranging from 7 to 35, with a higher total demonstrating greater hesitancy toward the COVID-19 vaccine.

## Acceptance of COVID-19 vaccine (AV)

The AV had two choices (1: yes, 0: no). The purpose was to collect evidence on the tolerability of COVID-19 inoculation. For example, "Do you accept to be vaccinated against COVID-19". If the answer 'yes' was selected, a choice of vaccine types was available in Thailand. These included 2 doses of inactive types (Sinovac and Sinopharm), 2 doses of adenovirus-based (AstraZeneca), 2 doses of mRNA-based vaccine (Pfizer and Moderna) and Cocktail (mixed type).

All questionnaire was written in the Thai language and the generated items were evaluated for content validity. Three experts as one doctor, one nurse and one researcher with extensive experience in the area of health literacy were invited to review the questionnaire for content validity. Content rationality determines whether the content of a scale is capable of calculating what it is planned to satisfy the research objective. Content validity and reliability were first tested for 30 participants and assessed using the index of item objective congruence (IOC). The IOC value was more than 0.7 for the whole questionnaire, while Cronbach's alpha coefficients were 0.85, 0.86 and 0.85 for VL, VH and VF, respectively. Mean scores and standard deviations (SD) of 5,312 VHVs were VL 33.83 (SD: 7.55), VF 11.35 (SD: 4.58) and VH 24.72 (SD:7.45), with Cronbach's alpha coefficients for VL, VH and VF 0.89, 0.91 and 0.90, respectively.

## Ethical approval and consent to participate

The study was reviewed and approved by the Institutional Ethical Review Board of the Faculty of Public Health, Chiang Mai University (IRB No.ET033/2021). Consent from the participants was obtained by asking them to sign an informed consent form by clicking on it. No animals were used in this research. All human research procedures were in accordance with the ethical standards of the committee responsible for human experimentation (institutional and national) following the Helsinki Declaration of 1975, revised in 2013.

## Statistical analysis

Definite facts were stated as figures and proportions, while incessant facts were summarized as means with standard deviations or medians with ranges (minimum: maximum) when continuous variables showed non-normal distribution.

The associations between VL, VF, VH, various demographic variables and AV were examined using logistic regression, with the outcomes stated as crude odds ratio (OR) and adjusted odds ratio (aOR) at 95%CI. VL, VF, and VH score were summed up individually and treated in models as a continuous independent variable. Significant features in the univariate analysis were employed as candidate variables in the initial model of multivariable analysis using backward elimination. The associations between VL, VF, VH, various demographic variables and AV were also examined by vaccine preference types in separate models using both univariate and multivariable analyses. A random effect by province was expected; however, low intraclass correlation coefficient results were recorded (ICC: 0.0535), and random effects were not assessed in this study. Statistical significance was set at $p < 0.05$. All examinations were conducted using Stata version 15.

## Results

### Demographic features

A total of 5,312 Thai VHVs completed the questionnaire via Google Form. Most VHVs who responded to our study were female (84.19%), aged 50–60 years old (36.09%), with work experience of less than 10 years (47.27%). Approximately 50% of VHVs had a high school education, a career in agriculture and no comorbidity. Most had an average monthly income of less than 10,000 baht and were couples (73.89%) and Buddhist (84.58%). Median VHV scores of VL, VF and VH were 2.92, 2.40 and 3.43, respectively.

Of the VHVs who accepted vaccination, a higher proportion (column percentage) preferred mRNA (56.52%), were female (85.22%), aged 50–60 years old (36.31%), had a high school education (54.34%), were couples (74.16%), Buddhist (86.09%), had a career in agricultural

(46.88%), had no comorbidity (60.51%), had average monthly income of less than 10,000 baht (45.66%) and work experience of less than 10 years (45.66) (Table 1).

**Effect of literacy, fear and hesitancy scores on recognition of COVID-19 inoculation.**
Table 2 shows the literacy, fear and hesitancy scores for AV. The univariate analysis results showed that each incremental unit of VL score significantly increased AV by 1.4% (OR = 1.014; 95%CI: 1.006–1.021), while, by contrast, fear and hesitancy scores significantly decreased AV by 4.4% (OR = 0.956; 95%CI: 0.944–0.967) and 3.7% (OR = 0.963; 95%CI: 0.954–0.971), respectively. After adjustment by variables in the multivariable analysis including sex, age group, education, religion, income and work experience, the VL score had a nonsignificant but modest effect on AV (aOR = 1.002; 95%CI: 0.994–1.01), while VF and VH significantly decreased the chance of AV (aOR = 0.966; 95%CI: 0.953–0.978) and 0.969 (0.960–0.979). Age group (40–50 years old vs <40 years old), education (high school, vocational certificate or higher level vs illiterate), religion (Islam and Buddhist), income (≥ 10,000 baht vs < 10,000 baht) and work experience (≥10 < 20 years, ≥20 years vs <10 years) were significantly associated with AV in the multivariable model.

**Association of literacy, fear and hesitancy scores on acceptance of COVID-19 vaccination by preference type.** The impact of VL score on AV differed for vaccine preference type (Table 3). The VL score showed a significant reverse effect on AV among VHVs who preferred mRNA vaccine (aOR = 0.984; 95%CI: 0.969–0.998), while VL scores increased the chance of AV in the remaining types of vaccines but not significantly. VF score was significantly associated with reduction of AV in VHVs who expected inoculation with inactive and adenovirus vaccines (aOR = 0.933; 95%CI: 0.901–0.966) and (0.917; 95%CI: 0.899–0.935), respectively. VH scores had a significant reverse effect on AV for most preference vaccine type in VHVs. The aOR values were 0.937 (95%CI: 0.912–0.963) for inactive vaccine, 0.947 (95%CI: 0.933–0.961) for adenovirus vaccine and 0.963 (95%CI: 0.945–0.982) for mRNA vaccine. The effects on other variables adjusted in the multivariable model by vaccine preference type are shown in the (S1 Table).

## Discussion

Several studies have investigated influencing factors connected with the reception of COVID-19 vaccination; however, few considered VL, VF and VH as predictors for AV, while none were conducted among VHVs in Thailand. Therefore, this study investigated the effect of VL, VF and VH on rapid contagious disease vaccination acceptance. The acceptance rate among Thai VHVs was 58.6%. After adjustment for demographic variables, VF and VH were significantly associated with decreasing VA, while VL was not significant in increasing vaccination of VHVs.

The AV rate of VHVs was moderate compared with the general population in other countries, ranging from 37.4% to 90% [10–14]. Vaccination acceptance was lower than among healthcare workers. A previous study of physicians in a university-based teaching hospital in Thailand found that better VL was inversely related to VH (aOR 0.34; 95% CI 0.13–0.9; p = 0.029) [22]. Parents with higher VL preferred to vaccinate their children compared to those with lower VL [29]. This is implying the role of VL in reducing VH and increasing AV. In the univariate analysis of our study, VHVs with high VL had stronger AV. There has been considerable discussion about the relationship between VL and the adoption of the COVID-19 vaccine. Numerous studies have suggested that high levels of VL contribute to AV. People with low VL have difficulty accessing health information, leading to poor vaccine decision-making. Individuals with high VL are able to access, comprehend, analyze, assess and disseminate vaccine information to others [30]. Therefore, encouraging individuals to have high levels of

**Table 1. Demographic, literacy, fear and hesitancy vaccine scores of Thai village health volunteers.**

| Variable | Vaccinated n (%) | Unvaccinated n (%) |
|---|---|---|
| **Preferred type of vaccine** | | |
| None | - | 2200 (100) |
| Inactivated | 503 (16.16) | - |
| Adenovirus | 665 (21.37) | - |
| mRNA | 1759 (56.52) | - |
| Cocktail | 59 (1.90) | - |
| Any | 126 (4.05) | - |
| **Sex** | | |
| Male | 460 (14.78) | 380 (17.27) |
| Female | 2652 (85.22) | 1820 (82.73) |
| **Age** | | |
| <40 | 472 (15.17) | 377 (17.14) |
| 40–50 | 989 (31.78) | 608 (27.64) |
| 50–60 | 1130 (36.31) | 787 (35.77) |
| ≥ 60 | 521 (16.74) | 428 (19.45) |
| **Education** | | |
| Illiterate | 12 (0.39) | 19 (0.86) |
| Elementary School | 1020 (32.78) | 940 (42.73) |
| High School | 1691 (54.34) | 1038 (47.18) |
| Vocational Certificate | 389 (12.50) | 203 (9.23) |
| **Marital status** | | |
| Single, widowed | 804 (25.84) | 583 (26.50) |
| Couple | 2308 (74.16) | 1617 (73.50) |
| **Religion** | | - |
| Buddhist | 2679 (86.09) | 1814 (82.45) |
| Christian | 54 (1.74) | 34 (1.55) |
| Islam | 379 (16.00) | 352 (12.18) |
| **Occupation** | | |
| Agriculture | 1459 (46.88) | 1004 (45.64) |
| Own business | 533 (17.13) | 380 (17.27) |
| Freelancer | 826 (26.54) | 578 (26.27) |
| Government officer | 26 (0.84) | 13 (0.59) |
| Private employee | 32 (1.03) | 29 (1.32) |
| Unemployed | 236 (7.58) | 196 (8.91) |
| **Comorbidity** | | |
| None | 1883 (60.51) | 1275 (57.95) |
| Diabetes | 239 (7.68) | 228 (10.36) |
| Hypertension | 452 (14.52) | 295 (13.41) |
| Hyperlipidemia | 129 (4.15) | 90 (4.09) |
| Obesity | 61 (1.96) | 56 (2.55) |
| Bone and skeletal disorder | 64 (2.06) | 56 (2.55) |
| Other | 284 (9.13) | 200 (9.09) |
| **Income per month** | | |
| < 10,000 baht | 2173 (69.83) | 1722 (78.27) |
| ≥ 10,000 baht | 939 (30.17) | 478 (21.73) |
| **Work experience** | | |
| <10 years | 1421 (45.66) | 1090 (49.55) |

*(Continued)*

**Table 1.** (Continued)

| Variable | Vaccinated n (%) | Unvaccinated n (%) |
|---|---|---|
| $\geq$10 < 20 years | 1031 (33.13) | 618 (28.09) |
| $\geq$20 years | 660 (21.21) | 492 (22.36) |
| Vaccine literacy score median (IQR) | 2.92 (0.75) | |
| Vaccine fear score median (IQR) | 2.40 (1.60) | |
| Vaccine hesitancy score median (IQR) | 3.43 (1.57) | |

overall health literacy and VL would have a positive effect and improve access to and use of health information. Thus, understanding overall health literacy, as well as VL, is important for specific immunization situations [31]. However, analysis of variables in the multivariable analysis indicated that VL was not statistically significantly related to AV, while AV was influenced by other factors apart from VL [17]. After adjustment with variables in the multivariable analysis, the VL score was non-significantly positively associated with AV, implying that AV was influenced by other factors and not VL per se.

We also investigated the association between VH and AV among VHVs and found a significant negative association between hesitancy and acceptance, implying that an increase in VH may reduce AV. VH is a long-standing phenomenon that poses a severe threat to global health and some infectious illnesses have recently resurfaced [32]. The WHO defines vaccination apprehension as 'delay in acceptance or refusal of vaccination despite the availability of vaccination services', while vaccine acceptance refers to the likeliness to get vaccinated [33, 34]. VH has long been a serious global issue [35], while COVID-19 inoculation indecision may be the tip of the iceberg of overall serum uncertainty in Thailand.

Among VHVs, higher fear levels were related to lower acceptance of COVID-19 inoculation. VF was the foremost cause of non-acceptance and had an undesirable impact on COVID-19 inoculation recognition in line with other previous findings [36, 37]. Fear is defined as an unfriendly expressive state produced by the insight of a threatening incentive [38]. As a result, increasing epidemic length heightens the qualms of the public and impacts their happiness and psychological health [38, 39]. Several studies reported that more cultured and knowledgeable people suffered less distress from COVID-19, highlighting the necessity of teaching and transparent public health policies. VHVs are intermediate communicators between healthcare professionals and people in the community; thus, the studying findings are generalized and limited to these groups.

The impact of VL, VF and VH on AV was further investigated for preferred vaccine types available in Thailand as inactive, adenovirus, mRNA and cocktail. Previous studies suggested that VL increased AV, [40, 41]. Our results indicated that VL insignificantly increased the chance of vaccination with inactive (OR = 1.002; 95%CI: 0.982–1.022), adenovirus (aOR = 1.011; 95%CI: 0.999–1.022) and Cocktail types (aOR = 1.008; 95%CI: 0.994–1.021). VL had a borderline impact on VHVs who wished to be vaccinated with the adenovirus-based type. The adenovirus has long been developed, manufactured and used in the real world for preventing diseases such as highly pathogenic avian influenza and Ebola [42–44]. The adenovirus vaccine showed an acceptable efficacy of 76%, higher than the inactive type (Sinovac had an efficacy of 51%) [45]. Hence, increasing VL may lead VHVs to seek, judge and decide to choose adenovirus. The effect of VL was reversely significant on those who preferred mRNA type, while the mRNA vaccine was associated with an increase in VH (aOR 8.86; 95% CI 1.1–71.54; p = 0.041) [22]. The mRNA showed promising efficacy in COVID-19 prevention and is comparatively new compared with the inactive and adenovirus-based types. In the past in

**Table 2. Effect of literacy, fear and hesitancy scores on acceptance of COVID-19 vaccine.**

| Variable | Crude OR | 95%CI | Adjusted OR | 95%CI |
|---|---|---|---|---|
| **Vaccine literacy score** | 1.014 | 1.006–1.021 | 1.002 | 0.994–1.010 |
| **Vaccine fear score** | 0.956 | 0.944–0.967 | 0.966 | 0.953–0.978 |
| **Vaccine hesitancy score** | 0.963 | 0.954–0.971 | 0.969 | 0.960–0.979 |
| **Sex** | | | | |
| Male | 1 | | 1 | |
| Female | 1.204 | 1.037–1.396 | 1.123 | 0.963–1.309 |
| **Age group** | | | | |
| <40 | 1 | | 1 | |
| 40–50 | 1.299 | 1.097–1.538 | 1.288 | 1.076–1.541 |
| 50–60 | 1.147 | 0.974–1.350 | 1.168 | 0.970–1.407 |
| ≥ 60 | 0.972 | 0.807–1.171 | 1.001 | 0.805–1.246 |
| **Education** | | | | |
| Illiterate | 1 | | 1 | |
| Elementary School | 1.718 | 0.829–3.558 | 1.427 | 0.675–3.017 |
| High School | 2.579 | 1.246–5.335 | 2.145 | 1.015–4.535 |
| Vocational Certificate | 3.034 | 1.444–6.374 | 2.404 | 1.118–5.165 |
| **Marital status** | | | | |
| Single, widowed | 1 | | - | - |
| Couple | 1.035 | 0.914–1.171 | - | - |
| **Religion** | | | | |
| Buddhist | 1 | | 1 | |
| Christian | 1.075 | 0.697–1.658 | 1.130 | 0.723–1.765 |
| Islam | 0.729 | 0.623–0.852 | 0.571 | 0.688–0.797 |
| **Occupation** | | | | |
| Agriculture | 1 | | - | - |
| Own business | 0.965 | 0.827–1.126 | - | - |
| Freelancer | 0.983 | 0.860–1.123 | - | - |
| Government officer | 1.376 | 0.703–2.691 | - | - |
| Private employee | 0.759 | 0.456–1.263 | - | - |
| Unemployed | 0.829 | 0.674–1.017 | - | - |
| **Comorbidity** | | | | |
| None | 1 | | - | - |
| Diabetes | 0.710 | 0.584–0.862 | - | - |
| Hypertension | 1.037 | 0.881–1.221 | - | - |
| Hyperlipidemia | 0.971 | 0.734–1.282 | - | - |
| Obesity | 0.738 | 0.509–1.067 | - | - |
| Bone and skeletal disorder | 0.774 | 0.536–1.115 | - | - |
| Other | 0.961 | 0.791–1.167 | - | - |
| **Income per month** | | | | |
| < 10,000 baht | 1 | | 1 | |
| ≥ 10,000 baht | 1.557 | 1.371–1.767 | 1.385 | 1.213–1.581 |
| **Work experience** | | | | |
| <10 years | 1 | | 1 | |
| ≥10 < 20 years | 1.280 | 1.126–1.453 | 1.252 | 1.092–1.434 |
| ≥20 years | 1.029 | 0.893–1.184 | 1.030 | 0.899–1.233 |

-: variable did not reach statistical significance in univariate analysis and was not included in the multivariable model

**Table 3. Effect of literacy, fear and hesitancy scores on vaccine acceptance by preference type.**

| | Vaccine types | | | | | | | |
| --- | --- | --- | --- | --- | --- | --- | --- | --- |
| | Inactive [a] | | Adenovirus [b] | | mRNA [c] | | Cocktail [d] | |
| | aOR | 95%CI | aOR | 95%CI | aOR | 95%CI | aOR | 95%CI |
| Vaccine literacy score | 1.000 | 0.980–1.020 | 1.010 | 0.994–1.018 | 0.984 | 0.969–0.998 | 1.013 | 1.000–1.027 |
| Vaccine fear score | 0.933 | 0.901–0.966 | 0.917 | 0.899–0.935 | 0.990 | 0.965–1.016 | 0.993 | 0.972–1.014 |
| Vaccine hesitancy score | 0.937 | 0.912–0.963 | 0.947 | 0.933–0.961 | 0.963 | 0.945–0.982 | 0.986 | 0.971–1.002 |

[a]: adjusted odds ratio (aOR) of literacy, fear and hesitancy scores for sex, age, occupation and comorbidity

[b]: adjusted odds ratio (aOR) of literacy, fear and hesitancy scores for sex, age, income and work experience

[c]: adjusted odds ratio (aOR) of literacy, fear and hesitancy scores for sex, age and income

[d]: adjusted odds ratio (aOR) of literacy, fear and hesitancy scores for sex, age, religion, comorbidity and income

Thailand, fake news about the safety or efficacy of vaccines has been released with details unconfirmed by experts [46]. This has impacted the trust and acceptance of vaccines by some Thai people. Those with higher VL may have the ability to source information and decide to not select mRNA, instead choosing previously demonstrated vaccine types such as adenovirus-based.

Both VF and VH were associated with acceptance or willingness to receive the vaccination [47–50], while broadly based studies revealed that fear of the disease encouraged vaccination [48, 51, 52]. Our team also investigated the impact of VF and VH on AV. Results showed that both VF and VH were associated with decreasing AV in all types of vaccines. VF and VH caused vaccination refusal from the fear of adverse side effects, safety and efficacy concerns and the short duration of clinical trials, with more information desired on vaccine approval mechanisms [53, 54]. Other studies also reported that vaccine hesitancy and refusal occurred due to concerns about safety, general lack of trust and doubts about the efficiency and provenience of the vaccine [55, 56].

From the public health viewpoint, our study was conducted on Thai VHVs as intermediate mediators between government health professionals and the public. We believe that convincing this group of people will encourage people in the community to receive COVID-19 vaccines. Based on our findings, the government should not pay attention to the VL of VHVs but instead focus on minimizing VF and VH to convince VHVs to accept vaccination as worthwhile.

This study had certain limitations. First, our study was conducted among Thai VHVs, with results generalized to health volunteers or healthcare workers. Data were not collected among VHVs in all provinces, and did not take into account random effects by province; however, the study participants included representatives of each region in Thailand. Second, the participation rate was low compared to the total eligible number of VHVs because no compensation or rewards were offered to those who completed the questionnaire. Further studies should attract more samples by providing remuneration for those who completed the survey. Third, the study results did not reflect true vaccination numbers because vaccines available in Thailand were restricted to inactivated types. Therefore, AV was only related to preference vaccine types. Finally, online data collection relates to population samples and their non-random nature. The researchers had no control over who and how many people filled out the questionnaire. As a result, most of the samples were women and this caused the data to be skewed. In Thai society, men are the main income earners of the family, while women stay at home or work at home. Therefore, more women apply for work as VHVs to perform community healthcare duties, working mostly during the day.

## Conclusions

Our cross-sectional study revealed that VL may not be a factor contributing to the acceptance of COVID-19 vaccination among VHVs in Thailand. Increasing VL obstructed VHVs who accepted to be vaccinated with the mRNA-based vaccine. To boost their vaccination acceptance, the government or health-related departments should focus on reducing VF and VH in Thailand.

## Supporting information

**S1 Table. Effect of literacy, fear and hesitancy vaccine score on acceptance by COVID-19 vaccine types: Univariate and multivariable analysis.**
(DOCX)

**S1 Dataset.**
(XLSX)

## Acknowledgments

The authors would like to acknowledge the village health volunteer network in Thailand for distributing the online survey.

## Author Contributions

**Conceptualization:** Pallop Siewchaisakul, Pongdech Sarakarn.

**Data curation:** Sirinya Nanthanangkul, Jirapat Longkul.

**Formal analysis:** Pallop Siewchaisakul, Jukkrit Wungrath.

**Funding acquisition:** Pallop Siewchaisakul, Waraporn Boonchieng.

**Investigation:** Pallop Siewchaisakul, Jukkrit Wungrath.

**Methodology:** Pallop Siewchaisakul, Jukkrit Wungrath.

**Project administration:** Pallop Siewchaisakul.

**Software:** Pallop Siewchaisakul, Jukkrit Wungrath.

**Supervision:** Waraporn Boonchieng, Jukkrit Wungrath.

**Validation:** Jukkrit Wungrath.

**Writing – original draft:** Pallop Siewchaisakul, Jukkrit Wungrath.

**Writing – review & editing:** Pallop Siewchaisakul, Jukkrit Wungrath.

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
