## [Decision Letter · Decision Letter 0]

8 Mar 2022

PONE-D-22-04125Role of Literacy, Fear, Hesitancy on Acceptance of COVID-19 Vaccine among Village Health Volunteer in ThailandPLOS ONE

Dear Dr. Wungrath, 

Thank you for submitting your manuscript to PLOS ONE. After careful consideration, we feel that it has merit but does not fully meet PLOS ONE’s publication criteria as it currently stands. Therefore, we invite you to submit a revised version of the manuscript that addresses the points raised during the review process.

We look forward to receiving your revised manuscript.

Kind regards,

Wenping Gong, Ph.D.

Academic Editor

PLOS ONE

Journal Requirements:

" ext-link-type="uri" xlink:type="simple">https://journals.plos.org/plosone/s/file?id=ba62/PLOSOne_formatting_sample_title_authors_affiliations.pdf"

"This research is supported by the Center of Excellence in Community Health Informatics, Chiang Mai University. The authors would like to acknowledge the village health volunteer network in Thailand for distributing the online survey. "

"This study was granted by the Center of Excellence in Community Health Informatics, Chiang Mai University, Chiang Mai, Thailand."

Reviewers' comments:

Reviewer's Responses to Questions

**Comments to the Author**

1. Is the manuscript technically sound, and do the data support the conclusions?

Reviewer #1: Partly

2. Has the statistical analysis been performed appropriately and rigorously? 

Reviewer #1: No

3. Have the authors made all data underlying the findings in their manuscript fully available?

Reviewer #1: No

4. Is the manuscript presented in an intelligible fashion and written in standard English?

Reviewer #1: No

5. Review Comments to the Author

Reviewer #1: The authors use cross-sectional data collected from a large and diverse sample of Thai village health volunteers to determine what factors are associated with COVID vaccination intentions. Results indicate that vaccine hesitance and fear are both associated with lower likelihood of getting vaccinated. The theoretical contribution is small, but there is considerable value in studying health behaviors in non-WEIRD (western, educated, industrialized, rich and democratic) countries. However, the manuscript has important shortcomings. There are data limitations, analytic errors, and the writing is unacceptable (e.g., there are numerous grammatical errors). These problems appear to be fixable, though doing so will require considerable effort.

Given the numerous opportunities for improvement, I organize my review in the order in which I first observed each problem.

- The introduction is unfocused and often off topic. This manuscript promises to make a modest empirical contribution, documenting attitudes associated with vaccine acceptance in Thailand. The introduction should explain this succinctly, summarizing what is known about other part of the world and explaining why readers might expect Thailand to be different.

- The description of the sample (page 4 line 100-) and of the procedures for collecting data (page 6 line 163-) do not appear entirely consistent. For example, the former implies that participants were registered users of a mobile app called “SMART VHV.” The latter says that participants were recruited using a form of snowball sampling via social media (Facebook and Line). I encourage the authors to combine these two sections and to carefully edit the content for clarity and consistency.

- There are important details missing from the description of the sampling. For example, how many individuals were invited to participate? See the AAPOR guidelines for reporting survey data.

- The authors imply that the sample is diverse and largely unbiased, but descriptive feature (page 7 line 195) suggest that it is highly skews (e.g., 84% female)

- Figure 1 is missing

- The description of the measures (starting on page 5) appears to contain many errors. For example, how can a 12-item scale include items numbered 11-15? And how can the same item numbers be used in several scales?

- When describing a scale, the authors should also report the M, SD, and alpha.

- What the authors describe as vaccine literacy appears to tap several different ideas, include efficacy. If this is an established scale, a cite would help.

- What the authors describe as vaccine hesitancy includes questions about effectiveness and perceived importance. If this is an established scale, a cite would help.

- The authors reject the use of a random effect based on the ICC (page 7 line 189). They should report the ICC.

- In the text, the authors say that participants were asked to specify their preferred vaccine if they indicated willingness to be vaccinated (page 6 line 148). However, Table 1 reports vaccine preference among those who do not wish to be vaccinated.

- The occupation categories in Table 1 are not mutually exclusive (e.g., employee and private employee).

- If the authors want to demonstrate that the association between vaccine fear and acceptance is conditioned on the individual’s vaccine preference, it would be more appropriate to test whether preference moderates the influence of fear when estimating acceptance.

- The discussion is too long and covers too many topics (including several that we not mentioned in the results section). As with the introduction, the manuscript would be more compelling if the authors restricted their comments to their most important contributions.

=== Potential PLOS policy violations ===

The authors checked the box indicating that the data will publicly available, but then write, “The datasets used and /or analyzed during the current study are available from the corresponding author on reasonable request.”

Participants were not allowed to submit the questionnaire if they skipped questions (page 7 line 175). Every US-based IRB that I have worked with requires that participants be allowed to refuse to answer questions.

6. PLOS authors have the option to publish the peer review history of their article (what does this mean?). If published, this will include your full peer review and any attached files.

Reviewer #1: No

---

## [Author Response · Author response to Decision Letter 0]

18 Apr 2022

Thank you very much for your fruitful comments. We have revised the manuscript following your suggestions and comments. We hope our revision would satisfy your expectation. Please find our point-by-point response below.

Comments to the Author

Reviewer #1: The authors use cross-sectional data collected from a large and diverse sample of Thai village health volunteers to determine what factors are associated with COVID vaccination intentions. Results indicate that vaccine hesitance and fear are both associated with lower likelihood of getting vaccinated. The theoretical contribution is small, but there is considerable value in studying health behaviors in non-WEIRD (western, educated, industrialized, rich and democratic) countries. However, the manuscript has important shortcomings. There are data limitations, analytic errors, and the writing is unacceptable (e.g., there are numerous grammatical errors). These problems appear to be fixable, though doing so will require considerable effort.

Given the numerous opportunities for improvement, I organize my review in the order in which I first observed each problem.

1. The introduction is unfocused and often off topic. This manuscript promises to make a modest empirical contribution, documenting attitudes associated with vaccine acceptance in Thailand. The introduction should explain this succinctly, summarizing what is known about other part of the world and explaining why readers might expect Thailand to be different.

Ans: We have re-written the whole introduction following your suggestions. (Page 3, line 51-91)

2. The description of the sample (page 4 line 100-) and of the procedures for collecting data (page 6 line 163-) do not appear entirely consistent. For example, the former implies that participants were registered users of a mobile app called “SMART VHV.” The latter says that participants were recruited using a form of snowball sampling via social media (Facebook and Line). I encourage the authors to combine these two sections and to carefully edit the content for clarity and consistency.

Ans: We totally agreed. We have combined the two sections. We first described about the registered SMART VHV, for these user group use social media (Line and Facebook). After that we described how we inform and distribute our online questionnaire. (page 4, line: 91-112)

3. There are important details missing from the description of the sampling. For example, how many individuals were invited to participate? See the AAPOR guidelines for reporting survey data.

Ans: We have revised the manuscript stated that “A nonprobability sampling so called the snowball sampling was adopted based on the 137,782 registered SMART VHV in our study through the social platform (Line and Facebook) in that these two social media are the most popular in Thailand and numbers of VHV use it to communicate and coordinate with each other.” (Page 4, line: 100-102)

4. The authors imply that the sample is diverse and largely unbiased, but descriptive feature (page 7 line 195) suggest that it is highly skews (e.g., 84% female)

Ans: This research is online data collection, and we did not specify the respondent. We understand that it will cause the data to be skewed. We therefore consider it a limitation of the present studies. (page 15, line 318-323). Moreover, Thai society men need to be the main income earner of the family and have to work outside while women are at home or working at home. Therefore, it is more likely that women will apply for VHV, which is obliged to perform community health care duties, which work mostly during the day.

5. Figure 1 is missing

Ans: We have submitted the Fig 1 (separate file).

6. The description of the measures (starting on page 5) appears to contain many errors. For example, how can a 12-item scale include items numbered 11-15? And how can the same item numbers be used in several scales?

Ans: We sincerely apologize for misunderstood in the use of characters. We explain that we planned to utilize the letter l (lowercase l) for the questions in the Vaccine Literacy (VL), which resembles the number 1. So, for the questions in this category, we've changed it to capital L for clarification. (page 5, line 127, 129 and 131-132).

7.When describing a scale, the authors should also report the M, SD, and alpha.

 Ans: Thank you for your suggestions, we have reported the mean, SD and alpha as written in page 8 line 184-187. “Based on the present study, the mean score and standard deviation (SD) of VL was 33.83 (SD: 7.55), VF was 11.35 (SD: 4.58), and VH was 24.72 (SD:7.45). Cronbach’alpha coeffiecnt for VL, VH and VF were 0.89, 0.91 and 0.90, respectively.” (Page 6, line: 164-166)

8. What the authors describe as vaccine literacy appears to tap several different ideas, include efficacy. If this is an established scale, a cite would help.

Ans: The VL questionnaire was adapted from Vaccine Literacy Questionnaire-Italy (HlVa-IT) of Biasio LR. et al. (2020). We have added this reference in the Methods section. (Page 5, line 125-126)

9. What the authors describe as vaccine hesitancy includes questions about effectiveness and perceived importance. If this is an established scale, a cite would help.

Ans: The VH questionnaire was adapted from Hesitancy Scale (aVHS) of Akel HT. et al. (2021). We have added this reference in the Methods section. (Page 6 line: 144-145)

10. The authors reject the use of a random effect based on the ICC (page 7 line 189). They should report the ICC.

Ans: We have reported the ICC which was 0.053570. (page 7, line 186)

11. In the text, the authors say that participants were asked to specify their preferred vaccine if they indicated willingness to be vaccinated (page 6 line 148). However, Table 1 reports vaccine preference among those who do not wish to be vaccinated.

Ans: We have noted and corrected our mistaken of reporting number in table 1. (Page 9) 

PS: we also re-checked our results and have corrected the number report in Table 2 and Table 3 of adjusted OR.

12. The occupation categories in Table 1 are not mutually exclusive (e.g., employee and private employee).

Ans: Thank you very much for your comments. We have changed the word from employee to freelancer in all Tables.

13. If the authors want to demonstrate that the association between vaccine fear and acceptance is conditioned on the individual’s vaccine preference, it would be more appropriate to test whether preference moderates the influence of fear when estimating acceptance.

Ans: Thank you very much for your idea about testing of whether preference moderates the influence of fear when estimating acceptance. We did try to analyze but it is impossible to do so, since our questionnaire design was that for those who only chose accepted to get vaccine can later choose the preference types. Therefore, it cannot take the preference type as a covariate in the logistic model. 

 Accept to get shot

Preference type of vaccine No Yes

No 2,200 0

Inactivated 0 503

Adenovirus 0 665

mRNA 0 1,759

Cocktail 0 59

Any 0 126

Total 2,200 3,112

As a result, we decided to analyze the association between VL, VF, VH, and AV in different model based on each preference type of vaccine, adjusted by other variables. We have showed in our univariate, the effect of fear itself on acceptance by vaccine types were not different according to the Odd ratio in the univariate analysis.

14 The discussion is too long and covers too many topics (including several that we not mentioned in the results section). As with the introduction, the manuscript would be more compelling if the authors restricted their comments to their most important contributions.

 Ans: Thank you very much for your suggestions. We have revised our discussion follow our main aim and results. We arranged the discussion to follow first the result of factors associated with overall acceptance (Table 2) then factors associated with preferences types of vaccine (Table 3). Implication limitation and conclusion. (Page 12-15, line 234-330) 

=== Potential PLOS policy violations ===

The authors checked the box indicating that the data will publicly available, but then write, “The datasets used and /or analyzed during the current study are available from the corresponding author on reasonable request.”

Ans: All relevant data are within the paper and its Supporting Information files.

Participants were not allowed to submit the questionnaire if they skipped questions (page 7 line 175). Every US-based IRB that I have worked with requires that participants be allowed to refuse to answer questions.

Ans: We would like to extend our meaning of the sentence “Participants were not allowed to submit the questionnaire if they skipped questions” and make it clearer that is participants can allow to refuse to answer the question, or don’t participate, or stop doing the questionnaire at any time. There is a function of google form that you need to answer all question before submitting. We select this function, for we can protect an incomplete missing data. However, there are still have some variable or questions (We did not set up the function) that were incomplete as show in Figure 1.

---

## [Decision Letter · Decision Letter 1]

1 May 2022

PONE-D-22-04125R1Role of Literacy, Fear, Hesitancy on Acceptance of COVID-19 Vaccine among Village Health Volunteer in ThailandPLOS ONE

Dear Dr. Wungrath,

Thank you for submitting your manuscript to PLOS ONE. After careful consideration, we feel that it has merit but does not fully meet PLOS ONE’s publication criteria as it currently stands. Therefore, we invite you to submit a revised version of the manuscript that addresses the points raised during the review process.

We look forward to receiving your revised manuscript.

Kind regards,

Wenping Gong, Ph.D.

Academic Editor

PLOS ONE

Journal Requirements:

Reviewers' comments:

Reviewer's Responses to Questions

**Comments to the Author**

1. If the authors have adequately addressed your comments raised in a previous round of review and you feel that this manuscript is now acceptable for publication, you may indicate that here to bypass the “Comments to the Author” section, enter your conflict of interest statement in the “Confidential to Editor” section, and submit your "Accept" recommendation.

Reviewer #2: All comments have been addressed

Reviewer #3: (No Response)

2. Is the manuscript technically sound, and do the data support the conclusions?

Reviewer #2: Yes

Reviewer #3: Yes

3. Has the statistical analysis been performed appropriately and rigorously? 

Reviewer #2: Yes

Reviewer #3: Yes

4. Have the authors made all data underlying the findings in their manuscript fully available?

Reviewer #2: Yes

Reviewer #3: Yes

5. Is the manuscript presented in an intelligible fashion and written in standard English?

Reviewer #2: Yes

Reviewer #3: Yes

6. Review Comments to the Author

Reviewer #2: The authors have done a thorough revision of the manuscript based on previous comments. However, there are some parts still need to be revised further to make the manuscript better. Please find below a few comments for your consideration.

Introduction

Authors presented the vaccination coverage in Thailand as of October 2021. It would be better if the figures for the current vaccination coverage in the country is presented.

In the introduction, the authors presented the COVID-19 acceptance rates around the world. To make this information complete, the status in Africa could be added. For instance: https://journals.plos.org/plosone/article?id=10.1371/journal.pone.0260575

The introduction still needs to be revised further for grammatical and presentation errors.

Authors should include the month in 2021 when the first batch of the COVID-19 vaccines arrived Thailand.

Methods

“Questionnaires” should be rewritten as “questionnaire” all through the manuscript.

The authors have not stated the exclusion criteria used in the selection of the participants.

Authors should provide the citations/references of the extensive literature used in this study.

Authors did not explain how the literacy, fear, and hesitancy scores were calculated before being used in the univariate and regression analysis. Ideally, this explanation should appear under the statistical analysis section.

Results

In Table 1, the heading of the first row should be: Vaccinated n (%) Unvaccinated n (%)

Discussion

Has been improved upon.

Reviewer #3: Paper by Dr. Wungrath et al. treated social and psychological factors associated with acceptance of COVID-19 vaccine in Thailand. I would like to present comments to improve the manuscript.

[Major]

1. Introduction section may be relatively lengthy. To introduce readers to Methods and Results, shorter introduction may be effective.

2. As a reader, I would like the authors to explain vaccine literacy score, vaccine fear score, and vaccine hesitancy score, briefly.

3. Table 3: I am interested in the results of pooled analysis of varied types of vaccines.

4. Table 3: Showing the results of the results of non-adjusted analysis may increase the understanding of this study results.

[Minor]

5. Many abbreviations may interrupt readers to smoothly read the manuscript. Key abbreviations could be spelled out.

Overall, something is missing in this paper. It lacks impact. But it is well written and deserves to be published.

7. PLOS authors have the option to publish the peer review history of their article (what does this mean?). If published, this will include your full peer review and any attached files.

Reviewer #2: **Yes: **Ismail A. Odetokun (Ph.D.)

Reviewer #3: No

---

## [Author Response · Author response to Decision Letter 1]

3 May 2022

Thank you very much for your fruitful comments. We have revised the manuscript following your suggestions and comments. We hope our revision would satisfy your expectation. Please find our point-by-point response below.

Review Comments to the Author

Reviewer #2: 

The authors have done a thorough revision of the manuscript based on previous comments. However, there are some parts still need to be revised further to make the manuscript better. Please find below a few comments for your consideration.

Introduction

1. Authors presented the vaccination coverage in Thailand as of October 2021. It would be better if the figures for the current vaccination coverage in the country is presented.

Ans: We have stated the current vaccination coverage in May 2022 is 70.3%. (Page 3; line: 58-59)

2. In the introduction, the authors presented the COVID-19 acceptance rates around the world. To make this information complete, the status in Africa could be added. For instance: https://journals.plos.org/plosone/article?id=10.1371/journal.pone.0260575

Ans: We have added the recommended reference as in the sentence “Vaccine acceptance rate was 37.40% in Jordan, 61.16% in Bangladesh, 56.90% in the EU, 80.00% in the USA and 63% in Africa [10-14]”. (Page 3; line: 62) 

3. The introduction still needs to be revised further for grammatical and presentation errors.

Authors should include the month in 2021 when the first batch of the COVID-19 vaccines arrived Thailand.

Ans: The first COVID-19 vaccine arrived in Thailand in February 2021 and the Thai Prime Minister encouraged VHVs to boost public confidence in this vaccine. Later, novel variants of COVID-19 and new vaccines arrived in Thailand (Page 4; line: 81)

Methods

4. “Questionnaires” should be rewritten as “questionnaire” all through the manuscript.

Ans: We have rewritten from questionnaires to questionnaire.

5. The authors have not stated the exclusion criteria used in the selection of the participants.

Ans: SMART VHVs who lack of address information and incomplete filling questionnaire were excluded. (Page 4-5; line 110-111)

6. Authors should provide the citations/references of the extensive literature used in this study.

Ans: We have provided citations/references of literature as recommendation. 

7. Authors did not explain how the literacy, fear, and hesitancy scores were calculated before being used in the univariate and regression analysis. Ideally, this explanation should appear under the statistical analysis section.

Ans: VL, VF, and VH score were summed up individually and treated in models as a continuous independent variable. (Page 7, line: 181-182)

Results

8. In Table 1, the heading of the first row should be: Vaccinated n (%) Unvaccinated n (%)

Ans: We have revised accordingly. 

Discussion

Has been improved upon.

 

Reviewer #3:

Paper by Dr. Wungrath et al. treated social and psychological factors associated with acceptance of COVID-19 vaccine in Thailand. I would like to present comments to improve the manuscript.

[Major]

1. Introduction section may be relatively lengthy. To introduce readers to Methods and Results, shorter introduction may be effective.

Ans: We shorten the introduction part as recommend.

2. As a reader, I would like the authors to explain vaccine literacy score, vaccine fear score, and vaccine hesitancy score, briefly.

Ans: We have briefly explained all scores as mentioned in the methods part. (Page 5-6; line: 124-150)

3. Table 3: I am interested in the results of pooled analysis of varied types of vaccines.

Ans: The pooled analysis result (all type of vaccines) have been shown in Table 2.

4. Table 3: Showing the results of the results of non-adjusted analysis may increase the understanding of this study results.

Ans: Thank you very much for your suggestions. We totally agree. However, we also concern about containing too many tables. Therefore, we put the overall results, showing both non-adjusted and adjusted results in the manuscript (Table 2). We provide the non-adjusted and adjusted full results varied by vaccine types in supporting information (S1 Table). 

[Minor]

5. Many abbreviations may interrupt readers to smoothly read the manuscript. Key abbreviations could be spelled out.

Ans: We have reduced some abbreviations. However, according to the journal regulation about abbreviations stated that “Do not use non-standard abbreviations unless they appear at least three times in the text”. Therefore, we keep other key abbreviations in the manuscript.

---

## [Editor Report · Decision Letter 2]

2 Jun 2022

Role of Literacy, Fear and Hesitancy on Acceptance of COVID-19 Vaccine among Village Health Volunteers in Thailand

PONE-D-22-04125R2

Dear Dr. Jukkrit Wungrath,

We’re pleased to inform you that your manuscript has been judged scientifically suitable for publication and will be formally accepted for publication once it meets all outstanding technical requirements.

Kind regards,

Wenping Gong, Ph.D.

Academic Editor

PLOS ONE

---

## [Editor Report · Acceptance letter]

14 Jun 2022

PONE-D-22-04125R2 

Role of Literacy, Fear and Hesitancy on Acceptance of COVID-19 Vaccine among Village Health Volunteers in Thailand

Dear Dr. Wungrath:

I'm pleased to inform you that your manuscript has been deemed suitable for publication in PLOS ONE. Congratulations! Your manuscript is now with our production department. 

Kind regards, 

on behalf of

Dr. Wenping Gong 

Academic Editor

PLOS ONE